

# Unraveling the hydrology and sediment balance of an ungauged lake in the Sudano-Sahelian region of West Africa using remote sensing

Silvan Ragettli [1], Tabea Donauer [1,2], Peter Molnar [2], Ron Delnoije [3], and Tobias Siegfried [1,4]

[1]hydrosolutions Ltd.,Venusstrasse 29, 8050 Zurich, Switzerland
[2]ETH Zurich, Institute of Environmental Engineering, Zurich, Switzerland
[3]Caritas Switzerland, Bamako, Mali
[4]ETH Zurich, Institute of Science, Technology and Policy, Zurich,Switzerland

**Correspondence:** Silvan Ragettli (ragettli@hydrosolutions.ch)

**Abstract.**

The presence of ephemeral ponds and perennial lakes in the Sudano-Sahelian region of West Africa is strongly variable in space and time. Yet, they have important ecological functions and societies are reliant on their surface waters for their lives and livelihoods. It is essential to monitor and understand the dynamics of these lakes to assess past, present, and future water

resource changes. In this paper, we present an innovative approach to unravel the sediment and water balance of Lac Wégnia, a small ungauged lake in Mali near the capital of Bamako. The approach uses optical remote sensing data to identify the shoreline positions over a period of 22 years (2000-2021) and then attributes water surface heights (WSHs) to each observation using the lake bathymetry. The method represents a significant advancement over previous attempts to remotely monitor lakes in the West African drylands, since it considers not only changes in water depth to explain recent declining trends in lake

areas, but also changes in the storage capacity. We recognize silting at the tributaries to the lake, but overall, erosion processes are dominant and threaten the persistence of the lake because of a continuous decrease of the floor level at the outflow. This explains the decreasing trend in WSH even for the wet-season, in spite of positive rainfall patterns.

## 1 Introduction

In arid and semi-arid areas of West Africa, small reservoirs and natural lakes improve the reliability of water supplies for

livestock and humans and allow the diversification of agricultural activities at the locale scale (Fowe et al., 2015). Wetlands are important for biodiversity maintenance, ecosystem functioning and conservation (e.g. Brouwer et al., 2014). At the same time, water resources in Sub-Saharan West Africa are under increasing pressure due to climatic changes, population growth and land degradation (Leblanc et al., 2008; Favreau et al., 2009; Oyebande and Odunuga, 2013). Changes in land use and land cover can have unexpected consequences on the dynamics of surface waters. Famously, a phenomenon referred to as the "the

Sahelian paradox" led to an increase in surface water despite a general precipitation decline during the last decades of the 20th century. The phenomenon has been first reported for small watersheds in Burkina Faso by Albergel (1987), and later also for several other watersheds in the West African Sahel (Descroix et al., 2009; Amogu et al., 2010). Increasing runoff coefficients because of overall vegetation decay due to the erosion of shallow soils and long drought periods have been identified as one





of the main drivers of the seemingly paradoxical eco-hydrological changes (Dardel et al., 2014; Gal et al., 2017). In addition,
the Sahel has seen a tendency toward rising daily precipitation extremes (Frappart et al., 2009; Panthou et al., 2014), and
increasingly concentrated runoff (Gal et al., 2017). The combination of all these effects leads to both higher inflow (Gal et al.,
2016) and higher sediment input to Sahelian lakes. The first has been manifested by a spectacular increase in pond and lake
areas in the pastoral Sahel, such as the Gourma region in northern Mali (Gardelle et al., 2010; Gal et al., 2017), and the latter
by observations of increasing turbidity and suspended particulate matter in lakes and ponds of the western Sahel (Robert et al.,
2017).

While the water availability in some natural freshwater reservoirs in the arid and semi-arid regions of Sub-Saharan West
Africa benefited from these changes in precipitation and runoff patterns, other lakes have seen strong declines in surface area.
As such, Lake Chad is the most famous example. The surface area of Lake Chad decreased by more than 90 percent between
the 1960s and the 1980s (Pham-Duc et al., 2020; Mahmood and Jia, 2019; Gao et al., 2011). The shrinkage of Lake Chad has
been attributed to severe droughts and increased irrigation withdrawals (Coe and Foley, 2001). The lake split in two parts in
1972. Because the southern pool receives more than 95% of river inflow, the northern pool ran completely dry in the 1980s and
has hardly recovered since then (Lemoalle et al., 2012). Excess water spilling to the northern pool is not sufficient to maintain
a permanent free water surface and, even without irrigation, the current climatology does not favour a single lake (Gao et al.,
2011).

In the past decades, Lake Chad has become a symbol of ongoing climate change and thus has attracted a lot of research
attention. Meanwhile, hydrological change at thousands of small ephemeral ponds and perennial lakes in the region remains
to be investigated (Haas et al., 2009). Such lakes are often located in isolated regions with no road access during the rainy
season, and the inflows to the vast majority of these water bodies are completely ungauged (Fowe et al., 2015; Gal et al., 2016).
The methods developed and applied to investigate Lake Chad changes are not easily transferable to small and ungauged water
bodies. Most commonly, satellite radar and laser altimetry have been used for determining variations in water surface heights
in time (Crétaux and Birkett, 2006; Zhu et al., 2017; Buma et al., 2018; Pham-Duc et al., 2020), but the method is restricted to
large lakes ($>50$ km$^2$) due to the poor density of altimetry tracks and the low revisit times (Crétaux et al., 2016; Avisse et al.,
2017). The lack of inflow observations in small water bodies in the Sahel makes the calibration of hydrological models for
simulating the effects of human water use and climate variability on water storage and surface area (Gao et al., 2011) or for
assessing streamflow trends (Mahmood and Jia, 2019) difficult, if not impossible.

Among the few studies that unravelled the water balance of small water bodies ($<10$ km$^2$) in arid and semi-arid regions
of Sub-Saharan West Africa, Fowe et al. (2015) measured rainfall, evaporation, and reservoir water level at a small reservoir
in Burkina Faso for a two-year period. They concluded that available water resources in the studied system were adequate to
fulfill existing demands. Soti et al. (2010) presented the application of a simple hydrological model to 98 seasonal ponds in
North Senegal for water level simulations. The spatio-temporal dynamics of the ponds were successfully reproduced, but the
model required daily field data for calibration (rainfall, water level), and its performance strongly depended on the quality of
available rainfall inputs. Gal et al. (2016) has combined area-volume (AV) relationships of three small Sahelian lakes with
daily evaporation and precipitation data to estimate water inflow to the lakes. They succeeded to quantify the processes behind



the Sahelian paradox by showing that the ratio between annual water inflow and precipitation has increased in the last 60
years. The study by Gal et al. (2016) demonstrated that it is possible to unravel the hydrology of small Sahelian lakes without
in-situ measurements. However, deriving inflows by empirical AV scaling relationships disregards the sedimentation of natural
lakes. The Sahel region is marked by a high degree of weathering due to the climatic conditions (Nippes, 1984) and increasing
soil erosion due to land degradation and concentrated runoff (Karambiri et al., 2003; Amogu, 2009; Descroix et al., 2009).
Constant AV relationships are ignoring the fact that the reservoir capacity of Sahelian lakes may naturally change due to
sediment deposition and erosion.

The objectives of our study are: (1) to develop and test a new method for quantifying both the sediment and the water balance
of an ungauged lake (Lac Wégnia in Mali) based on remote sensing information, (2) to quantify the evolution of water surface
elevation and surface area over the last twenty-two years in the lake, and (3) to attribute possible causes to observed changes
and identify adequate measures to safeguard the ecological balance and environmental equilibrium of the lake.

The method we are proposing is based on the waterline method (Mason et al., 1995). The waterline (or shoreline) refers to
the water-land boundary and can be regarded as a contour line that connects points of equal elevation. The general methodology
consists of detecting the ever-shifting edge of water bodies in remotely sensed images using image processing techniques and
assigning heights to shoreline points using water level information or bathymetry data. The method is a widely used technique
for constructing intertidal DEMs (Salameh et al., 2019). Morphological change can then be quantitatively estimated based on
DEMs generated for different years or seasons (e.g. Mason et al., 1999; Ryu et al., 2008; Heygster et al., 2010; Li et al., 2014;
Xu et al., 2016). Only in recent years, likely because of better data coverage and availability of remote sensing imagery, the
method has been adapted to generate time-series of water surface heights (WSH) of lakes and reservoirs (Tseng et al., 2016;
Ma et al., 2019; Weekley and Li, 2019; Yue and Liu, 2019; Militino et al., 2020; Xu et al., 2020) or for Water volume estimates
of desert lakes Armon et al. (2020). The method is only applicable to recently filled reservoirs or other water bodies where the
water level has increased above the level at the time the elevation data were collected. The method has not yet been employed
in Sub-Saharan West Africa, despite the fact that the region is particularly suitable for an application of the method, because
bathymetric surveys can be carried out towards the end of the dry season, when the lake levels are at their lowest.

For the present work we apply the waterline method with a digital elevation model (DEM) derived from an unmanned aerial
vehicle (UAV) survey in May 2019. The shorelines of Lac Wégnia are identified by leveraging the Landsat and Sentinel-2
image archives. A novel methodology is presented to identify and quantitatively analyze deposition and erosion patterns at
the lake shores and within the lake bed by mapping the temporal evolution of shoreline position anomalies. Finally, reservoir
capacity changes and storage variations are retrieved, which allows us to carry out a detailed water balance analysis of Lac
Wégnia.

## 2 Study site and climate

Lac Wégnia (13°18'00"N, 08°07'46"W) is located in south-west Mali, approximately 75 km north of Bamakao (Figure 1), in
the watershed of the Senegal River and more specifically in the watershed of the Baoulé River, the main tributary of Bakoy





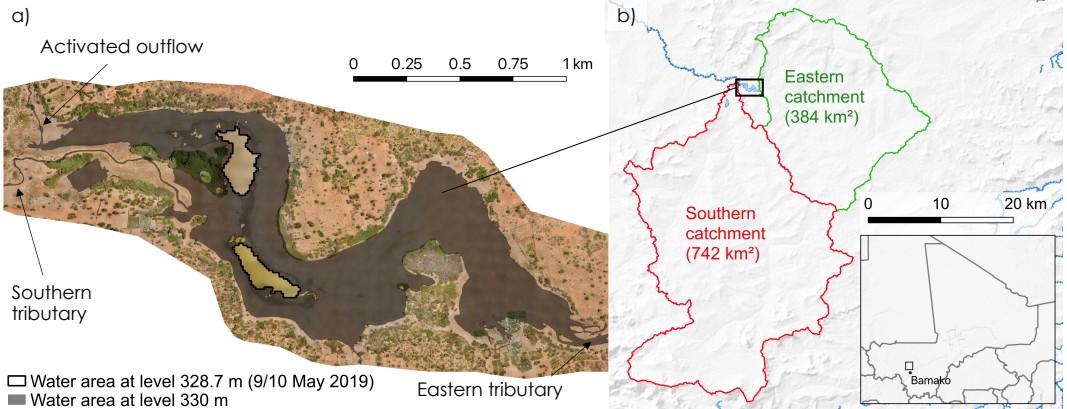

**Figure 1.** Overview map of Lac Wégnia. a) Ortho image of the drone footage from 9/10 May 2019. The water area at level 330 m asl (based on the UAV-DEM) and on the day of the drone flights (328.7 m asl) are shown. The outflow is activated if the water level reaches above 329.8 m asl. b) Outline of the two main contributing catchments and location of Lac Wégnia in Mali.

River. With an average annual rainfall of about 850 mm the lake is at the boundary between the Sahelian and the Sudanian eco-climate. The climate is characterized by convective heavy rainfalls during the wet season resulting from the West-African monsoon, and a long dry season. The rainy season is short, extending typically from late June to mid-September. The mor-

95 phology of the surrounding region is characterized by the predominance of sandstone plateaus often covered with ferruginous crusts between 300 and 400 m altitude (de La Rocque and Renoullin, 2015).

In 2013 Lac Wégnia was designated as a RAMSAR site (Coulibaly et al., 2013) and is thus part of the RAMSAR list of wetlands of international importance. The entire RAMSAR site has an area of 3,900 ha and includes also several smaller lakes and ponds in the vicinity of Lac Wégnia. The freshwater lakes and marshes of the Lac Wégnia RAMSAR site are unique in

the region for their ecological characteristics and natural state (Coulibaly et al., 2013). Lac Wégnia plays an essential role in the natural control of floods. The seasonal water retention by the lake is important for the wetlands and the entire surrounding area. About 12,000 people depend directly on the lake and its surroundings for food and for economic activities such as fishing, raising livestock and agriculture (DNEF/PAZU, 2018).

Lac Wégnia has two main tributaries (Figure 1), one from the south (catchment area 742 km$^2$) and one from the east (384

105 km$^2$). The entire catchment area of the lake is about 1157 km$^2$. The lake drains at its northwestern end through a narrow gully, but outflow from the lake is only activated during the wet season. During the entire dry season the river bed of the effluent and of the tributaries are dry. Towards the end of the wet season the lake extends to an area of about 1500 ha, but then continuously decreases in size during the dry season. While at the beginning of the century the lake rarely decreased to areas of less than 200 ha, this is now common. In recent years the surface water area has decreased to less than 10 ha in late May and early June (de

La Rocque and Renoullin, 2015). To safeguard the ecological balance, there is urgent need to understand the causes of recent water area trends and to assess the changes in lake water storage.





## 3   Data

### 3.1   UAV data

The UAVs eBee and RTK from SenseFly were used for the realization of the drone survey on 9/10 May 2019. The UAVs were
mounted with a senseFly S.O.D.A camera with a 20 megapixel sensor. Its lens was fixed at a focal length of 35 mm and the
altitude of the flights was 180 m. All the aerial photographs were processed using the commercial software Agisoft Photoscan
to generate the DEM. The orthomosaic images (Figure 1a) have a spatial resolution of 3.28 cm and the DEM a resolution of
6 cm. Based on three ground control points, the mean error in X and Y direction was estimated to be 1.6 cm and 5.6 cm in Z
direction, respectively. More technical details on the topographic survey are provided by Vandemeulebrouck et al. (2019).

### 3.2   Satellite remote sensing data

This research utilizes high spatial resolution remote sensing data from the following satellite missions: Landsat 5 (L5, 1984-
2012), Landsat 7 (L7, 1999-present), Landsat 8 (L8, 2013-present) and Sentinel-2 (2015-present). We process surface re-
flectance images from all available scenes in the L5, L7, L8 and S2 archive from the study period October 1999 - May 2021 in
Google Earth Engine (GEE, Gorelick et al., 2017).

### 3.3   Meteorological data

Meteorological data used in this study is provided by various gridded datasets (Table 1). We use all seven precipitation products
that are available for the study region through the Earth Engine Data Catalog. Two out of the seven products also provide the
necessary inputs for the calculation of evaporation rates from open water surfaces (Section 4.2: GLDAS (Rodell et al., 2004)
and ERA5 (Hersbach et al., 2020). All datasets are characterized by a high spatial (5-25 km) and temporal (hourly-daily)
resolution.

    In-situ observations are available from a station of the Trans-African Hydro-Meteorological Observatory (TAHMO) project
(van de Giesen et al., 2014) in Guioyo, approximately 25 km north-west of Lac Wégnia. We have set up the station in February
2020 within a fenced school compound. The station measures precipitation, air temperature, incoming shortwave radiation,
relative humidity, atmospheric pressure, wind speed and wind direction with the ATMOS41 automatic weather station from
METER. Wind data from the station are not used because of the anomalous wind conditions within the enclosed school
compound where the station was set up. Furthermore, due to prolonged dry periods and bad maintenance, likely related to
school closings during the Covid-19 pandemic, dust accumulated on the shortwave radiation sensor. Solar radiation data from
the station was therefore also not used.



**Table 1.** Meteorological datasets used in this study. Also indicated are wet season precipitation trends for Lac Wégnia that are significant at the 0.01 level (the values in brackets represent the precipitation change from 2000 to 2020 in percent). G: gauge; S: satellite; R: reanalysis; P: precipitation; $T_a$: air temperature; $T_d$: dewpoint temperature; W: wind, RAD: solar radiation, Qair: specific humidity; Pr: surface pressure; NP: near present; POS: positive trend; N: no trend.

| Name | GEE Code | Data Sources | Variables Used | Temporal Coverage | Temporal Resolution | Spatial Resolution | P Trend Wet Season |
|---|---|---|---|---|---|---|---|
| CHIRPS | UCSB-CHG/CHIRPS/DAILY | S, R, G | P | 1981-NP | daily | 0.05° | POS (+23.4%) |
| ERA5 | ECMWF/ERA5_LAND/HOURLY | R | P, $T_a$, $T_d$, W, RAD | 1981-NP | hourly | 0.25° | POS (+34.1%) |
| GLDAS | NASA/GLDAS/V021/NOAH/G025/T3H | S, G | P, $T_a$, W, RAD, Qair, Pr | 2000-NP | 3-Hourly | 0.25° | N |
| GPM | NASA/GPM_L3/IMERG_V06 | S, R, G | P | 2000-NP | 3-Hourly | 0.1° | N |
| GSMaP | JAXA/GPM_L3/GSMaP/v6/reanalysis | S, R, G | P | 2000-2013 | hourly | 0.1° | POS (+24.1%) |
| | JAXA/GPM_L3/GSMaP/v6/operational | S, R, G | P | 2014-NP | hourly | 0.1° | |
| PERSIANN-CDR | NOAA/PERSIANN-CDR | S, G | P | 1983-NP | daily | 0.25° | N |
| TRMM 3B42 | TRMM/3B42 | S, G | P | 1998-2019 | 3-Hourly | 0.25° | N |
| Ensemble-mean | | | P | 2000-2020 | | | POS (+21.2%) |

## 4 Methods

### 4.1 Lake water balance

The equation of lake level change over a given period is defined as follows (Eq. 1):

$$\Delta h = \sum_{i=t_1}^{t_2} (E_i + P_i + Q_i) \tag{1}$$

where $\Delta h$ is the change in WSH (mm) over a time period $t_1$ to $t_2$ and $E_i$, $P_i$ and $Q_i$ (mm/day) are the daily evaporation, precipitation and net inflow to the lake, respectively. The methodology to derive $\Delta h$, $E_i$ and $P_i$ is described below. In the absence of discharge measurements, $Q_i$ is used to close the water balance and includes all fluxes at the subsurface or surface, such as interactions with groundwater, surface water in- and outflow, and water losses due to human and animal consumption.

The water balance components are identified for 22 dryseasons between October 1999 to May 2021. The analysis focuses only on the loss part of the year to explain the ecologically most critical conditions for the lake. The wet-season water balance is not investigated in this work, also because of the higher uncertainties due to the absence of local precipitation measurements and the presence of clouds which reduce the availability of remote sensing imagery. The period of analysis was fixed from 1 October to 15 May (227 days) to ensure comparability between years.

### 4.2 Evaporation from open water

The Penman equation is commonly used in the literature to estimate evaporation from lake surfaces (e.g. Gal et al., 2016; Schulz et al., 2020). This equation is based on the energy balance and aerodynamic constraints. We use the 1948 version of the Penman model (Penman, 1948) implemented in the R package 'Evapotranspiration' (Guo et al., 2016). Open-water evaporation

Earth **Surface**
**Dynamics**
Discussions

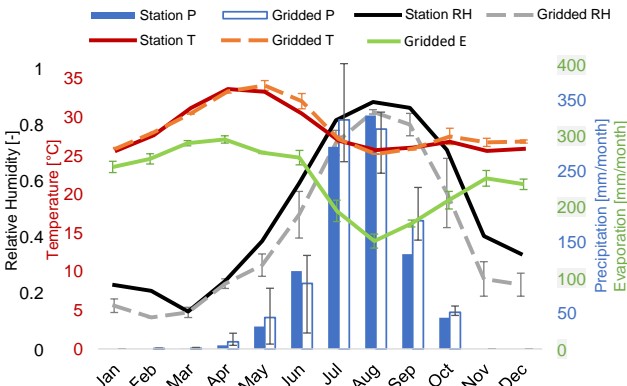

**Figure 2.** Climate diagram for Guioyo (located 25 km north-west of Lac Wégnia) with data from the period March 2020 - February 2021. 'Station' refers to the TAHMO station data and 'Gridded' refers the ensemble-mean of the gridded global datasets (Table 1). The error bars represent the full range of values among gridded datasets.

is estimated using the function ET.Penman(), whereas the arguments are set so that (1) the time-step for calculation is daily; (2) the Penman 1948 wind function (Penman, 1948) is used to estimate the mass transfer component in the Penman model that influences the rate of movement of water vapor away from the evaporating surface; and (3) the evaporative surface is open water (albedo = 0.08, roughness height = 0.001m). The required inputs are solar radiation ($RAD$), relative humidity ($RH$),

air-temperature ($T_a$) and wind speed ($W$). Due to the unavailability of nearby station data for the entire study period we assume that the weather data above the lake can be best approximated by the data from gridded products (Table 1).

For the present study these inputs are provided through the GLDAS and ERA5 datasets, respectively. The following ERA5 climate variables are used in the computation of evaporation: $T_a$, $RAD$, 2 m dew point temperature ($T_d$) and wind speed at 10 m above ground level. Relative humidity is calculated from $T_a$ and $T_d$ via the Magnus approximation (Alduchov and Eskridge,

1996). To compute GLDAS evaporation, the same variables are used except that relative humidity is converted from specific humidity, near surface air pressure and $T_a$. Each variable is validated against station data if available from the TAHMO station in Guioyo. The Penman model is then applied separately with inputs from each of the two datasets. The final time series of open-water evaporation is obtained by taking the arithmetic mean of the two datasets.

### 4.3 Precipitation

We use the gridded rainfall products (Table 1) to estimate precipitation at Lac Wégnia, and the TAHMO station data of 2020 for validation. It can be expected that the different gridded products have a substantially different performance in representing the spatio-temporal rainfall patterns (Dembélé et al., 2020). Here, we only assess the water balance of the dry period, where rainfall is scarce and where the different products generally agree well with each other (Figure 2). We therefore use the daily ensemble-mean of all 8 available products as an input for the precipitation component of the water balance.





### 4.4 Water surface heights (WSH)

WSH identification is developed below and implemented as a fully automated processing chain in GEE. The procedure involves the following main steps: i) remote sensing image selection and pre-processing, ii) water surface detection, iii) shoreline detection and iv) surface elevation retrieval.

#### 4.4.1 Remote sensing image pre-processing

The satellite images are co-registered with the Ortho image of the UAV survey in a step-wise manner. First, we selected an S2 image that was taken approximately on the same day as the UAV survey (11 May 2019). This image was co-registered with the Ortho image. We then assumed that the misalignment of images within the set of images from the same sensor is neglectable (Storey et al., 2016; Nguyen et al., 2020) and applied the same displacement to all S2 images. For the Landsat sensors no images were available from the date of the UAV survey. L7 and L8 images are thus co-registered with already aligned S2 images on the same day (25 April 2020 and 11 November 2020, respectively). The L5 and S2 mission periods do not overlap. In a final step L5 images are thus co-registered with a L7 image (9 June 2007). More details about co-registration in GEE are provided by Nguyen et al. (2020). All satellite images with a cloud percentage over the study area greater than 30% are excluded from the analysis.

#### 4.4.2 Water surface detection

Water absorbs most radiation at near-infrared wavelengths and beyond. As a result, water can be easily detected by using spectral indices. Here we use the modification of the normalized difference water index (MNDWI) for land-water classification (Xu, 2006):

$$MNDWI = (Green - MIR)/(Green + MIR) \qquad (2)$$

where $MIR$ is reflectance in the middle infrared band, and $Green$ is reflectance in the green band. The MNDWI has been extensively applied for water mapping (e.g. Donchyts et al., 2016; Tseng et al., 2016; Ma et al., 2019) and its good performance has been shown for both Landsat and Sentinel-2 images (Kwang et al., 2017; Yang et al., 2020).

We use a non-parametric unsupervised method based on the Canny edge filter and Otsu thresholding to distinguish between water and non-water pixels, following Donchyts et al. (2016). The Otsu algorithm (Otsu, 1979) is a widely-used dynamic threshold method (e.g. Ma et al., 2019; Asfaw et al., 2020; Yang et al., 2020). The method identifies an optimal threshold to distinguish two image classes by maximising the inter-class variance computed from a normalized image histogram. This requires a bimodal histogram distribution. If land dominates the image over water, the histograms are unbalanced. The Canny edge filter is thus used to reduce the sampling region only to those areas near water-land edges. The Canny method first uses a Gaussian filter to smooth the image in order to remove the noise and then finds edges by looking for local maxima of the image intensity gradients. We use the Canny edge algorithm with the following parameters to process all images: $\sigma = 0.7$ (standard





deviation of the Gaussian smoothing kernel), $th = 0.7$ (threshold used to define the sensitivity of the gradient magnitude filter), an image pixel resolution of 30 m and a buffer around the edges of 60 m.

Binary water images are obtained at the native resolution of the sensors (30 m for Landsat, 10 m for Sentinel-2). The area of Lac Wégnia can be obtained from these images after gap-filling. A simple focal mode filter is applied to fill data gaps and void stripes in Landsat 7 images. Note that gap filling is not required for WSH retrieval, which is another advantage of the method over AV scaling.

### 4.4.3 Shoreline detection and elevation retrieval

The shoreline is defined as the water/land boundary of the binary water image. To retrieve the elevation of the shoreline, a 10-m buffer is added on both sides of the edge. Within this buffer we sample all DEM pixel values at a 10-m resolution, and use the median to calculate a single WSH. Weekley and Li (2019) obtained slightly better results using the mean instead of the median. However, the median is less sensitive to noise and shoreline elevation anomalies (see Section 4.5).

A variety of factors such as local slope, mixed pixels, and water detection accuracy can impact the accuracy of the retrieved WSH (Tseng et al., 2016; Weekley and Li, 2019). Because surface water may not be correctly identified under canopies, we mask all boundary pixels that are adjacent to lush vegetation (NDVI > 0.5). Note that at Lac Wégnia this situation only occurs during the wet season, when the lake is completely filled. Pixels where the retrieved mean erosion/deposition rate exceeds 10 $mm/year$ are also removed from calculating the final WSH (see Section 4.5 below). Unfortunately, no in-situ water level data are available for Lac Wégnia for a validation of the retrieved WSH data. We therefore only assess the relative accuracy of the approach by comparing WSH data obtained for the same or subsequent days from different images (acquired by Landsat and Sentinel-2 satellites, respectively).

### 4.5 Sediment balance

The waterline method is based on the assumption of common heights of geocoded waterline pixels (Salameh et al., 2019). However, this assumption is only valid if either the elevation information is available exactly for the time of the lake boundary acquisition, or if the bathymetry of the lake does not change over time. In our case a DEM is only available for May 2019, and it is very likely that over the study period 1999-2021 there are changes in the lake lake shore and bed topography due to sediment deposition and erosion. Deposition occurs preferentially on sediment deposit cones, while lake shore erosion may take place where the banks are not protected by littoral vegetation and where rills and gullies are formed. Such local topographic changes become visible by comparing the waterlines of two scenes which represent approximately the same water level. At sediment deposition cones the waterline of the older scene bends much more inland than the more recent scene (Figure 3a), and vice versa at erosion zones. If such 'shoreline anomalies' are not too numerous, the median of all shoreline pixel elevations still provides a valid estimate of the water surface elevation, even if the lake boundary of an older scene is projected on the recent lake bathymetry. The difference between the median elevation of all shoreline pixels and a pixel elevation at a shoreline anomaly is then equal to the elevation change that has occurred at that point. In the following, we call these deviations from the median 'shoreline elevation anomalies' (SEAs):





$$SEA_{i,t} = WSH_t - SE_{i,t} \qquad\qquad\qquad (3)$$

$\text{SEA}_{i,t}$ in Eq. 3 is the SEA at time $t$ at pixel $i$, $\text{SE}_{i,t}$ is the shoreline elevation at pixel $i$ according to the current bathymetry,
and $\text{WSH}_t$ is the median of all shoreline pixel elevations. If a SEA is positive, then erosion has since occurred, and if it is
negative, deposition has occurred.

SEAs also occur if the scenes providing the lake boundaries are not well aligned with the DEM, or if the shoreline delineation
is erroneous. For this reason it is important to look at the evolution of SEAs over time to get a robust picture of the nature of
geomorphic change on the lake shoreline. SEAs should be continuously changing over time at deposition or erosion zones. Not
the SEAs per se point to deposition and erosion processes, but the slope of SEA change over time ($\Delta$SEA, units $mm/year$).
Robustly increasing or decreasing SEA slopes can then be assumed to be equivalent to sediment deposition or erosion rates,
respectively. Furthermore, SEAs may occur because of topographic slopes within the $\pm 10$m buffer that is used for sampling
shoreline pixels. For this reason, SEAs extracted at 10m pixel resolution are smoothed by applying a morphological mean filter
within 30m square kernels.

The complete procedure to extract maps of $\Delta$SEA can be summarized as follows: 1) Identifying the WSH of a given scene
as described in Section 4.4.3. 2) Mapping of SEAs within the $\pm 10$-m buffer of each lake boundary. 3) Smoothing of SEAs
with the morphological mean filter. 4) Taking the mean of all SEAs over a given pixel for a given hydrological year, resulting
in 22 annual SEA maps (hydrological years 2000-2021). 5) Applying a non-parametric Sen's slope estimator calculate $\Delta$SEA.
Sen's slope is less sensitive to outliers than the common least squares estimate using linear regression (Sen, 1968).

The validity of the procedure depends on the assumption that the median shoreline pixel elevation is a robust estimator of
actual WSH. To test this assumption, steps 1-5 above are applied iteratively, by always masking out in step 1 those areas where
absolute $\Delta$SEA exceeds 10 $mm/year$. Therefore, locations where anomalies occur get excluded from calculating WSH. After
each iteration the average $\Delta$SEA within the lake bed is calculated (hereafter referred to as the 'sediment balance'), as well as
the fraction of the area where absolute $\Delta$SEA does not exceed 10 $mm/year$. If both indicators converge to stable values after
10 iterations, the resulting $\Delta$SEA map can be used to identify the deposition and erosion areas that need to be masked for the
calculations of the final WSH values.

Because Lac Wégnia is not an ephemeral lake but shrinks to a small size each dry season, $\Delta$SEA cannot be calculated for
those lake bed pixels which are rarely or never dry. We apply a minimum threshold of six annual SEA values ( 30%) that need
to be available in order to calculate $\Delta$SEA. Data gaps in the map (Figure 3b) are filled with a focal mean filter (Figure 3c) if
pixels with $\Delta$SEA values are present within a two-pixel (20m) radius.

### 4.6 Trend analysis

We look for trends in the seasonal mean values of each water balance component (Eq. 1). Regression slopes are estimated using
the method of Theil (1950) and Sen (1968), thereafter called Sen's slopes, and uncertainty ranges are provided by the 95% CI





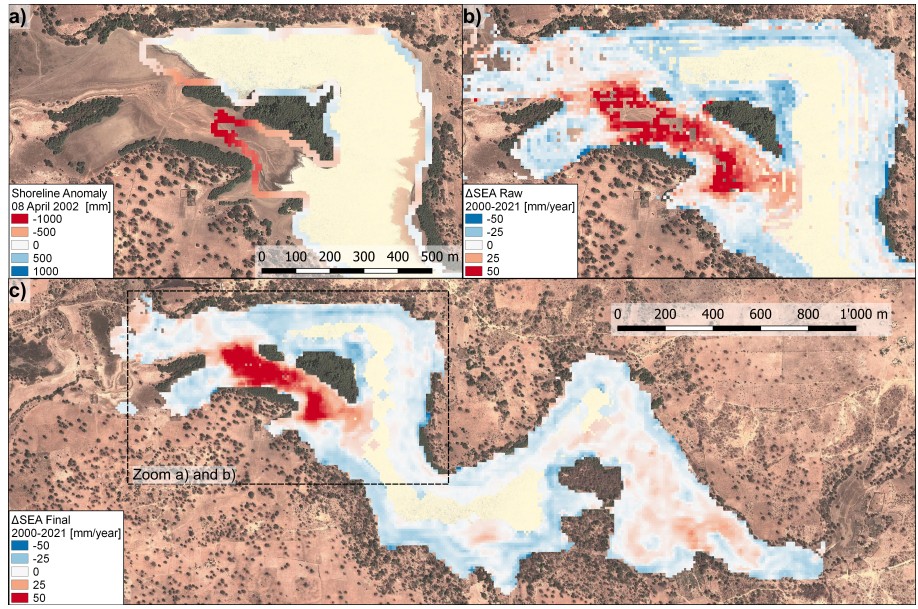

**Figure 3.** a) Shoreline elevation anomalies of a Landsat-7 scene from 8 April 2002. b) Slope of shoreline elevation anomalies (ΔSEA) 2000-2021 in meter per year. Shown are all all 10-m pixels where data points are available from at least six years. c) Gap filled and outlier corrected map of ΔSEA 2000-2021. Red and blue pixels point to sediment deposition and erosion, respectively. This map is used to calculate the sediment balance of Lac Wégnia. The background image is a Google Earth image from 23 March 2013 (© Google Earth 2021).

of the Sen's slope estimate computed using the Gilbert (1987) modification of the Theil/Sen Method. The significance of the
trend is assessed by Kendall's nonparametric test for a monotonic trend (Kendall, 1975).

To explain the observed lake level trends we perform an attribution analysis. Three main factors determine the lake level at
the end of the dry season: i) the initial lake level, ii) the mean rate at which the lake level decreases over the dry season and iii)
the length of the dry season. Changes in these three independent factors over the last 22 years can explain the observed lake
level trends:

$$\Delta h_{tot} = \Delta h_{rate} + \Delta h_{ini} + \Delta h_{\Delta t} \approx \Delta h_{obs} \tag{4}$$

whereas $\Delta h_{rate}$ represents the lake level change that is due to the change in the rate at which the lake level decreases over
the dry-season. $\Delta h_{ini}$ is the change in initial lake levels, and $\Delta h_{\Delta t}$ represents the lake level change due to changes in the
timing of the wet-season between. The sum of all effects ($\Delta h_{tot}$) should be approximately equal to the observed difference in
dry-season lake levels ($\Delta h_{obs}$). Each factor is quantified based on the available WSH time-series as explained in Sections 4.6.1
- 4.6.3 below.





### 4.6.1 Dry-season lake level change rates

The number of available WSH data points varies between years due to clouds and dust storms and due to the different mission periods of the satellites. To enable comparison of WSH change rates it is therefore necessary to homogenize the WSH time series. For each hydrological year, the available data are extrapolated to cover the full dry-season period from 1 October to

285 15 May. The steps for extrapolation of the data are the following: 1) Smoothing the WSH time-series with a 7-day window running-mean filter. 2) Calculation of the WSH change rate ($\Delta h$, units $mm/day$) between all subsequent dry-season data points that are separated by 8-16 days (16 days is the usual Landsat overpass interval). 3) The day of the year (DOY) of the last day of each interval is attributed to each available data point. The cloud of points is converted into a time series of average $\Delta h$ seasonality by smoothing again with a 7-day window running-mean filter. 4) Data gaps are filled by using the average $\Delta h$

seasonality as follows:

$$\Delta h_{gap} = \sum_{i=t_1}^{t_2} \Delta h_{ref_i} \times \frac{\Delta h_{obs_{t3-t4}}}{\sum_{i=t_3}^{t_4} \Delta h_{ref_i}} \tag{5}$$

where $\Delta h_{gap}$ is the interpolated WSH change (units in $mm$) for a given time period of missing data between $t_1$ to $t_2$. $\Delta h_{ref_i}$ is the reference WSH change (units in $mm/day$) described by the average $\Delta h$ seasonality for a given DOY, and $\Delta h_{obs_{t3-t4}}$ is the observed WSH difference between $t_3$ to $t_4$. The time period $t_3$ to $t_4$ has to cover at least 50% of the entire period 1

October to 15 May. Hydrological years for which less than 50% of the dry period is covered by available WSH observations are not considered for the $\Delta h_{rate}$ trend analysis. For all other hydrological years, an average $\Delta h_{rate}$ is calculated based on the gap filled dry-season WSH difference. The Sen's slope test is applied to assess the multi-year trend in $\Delta h_{rate}$. The gap filled dry-season WSH difference is also used together with $E$ and $P$ to derive $\Delta Q$ for each hydrological year based on Eq. 1.

### 4.6.2 Initial lake levels

During the wetseason, when the lake is completely full, the lake level raises above the elevation of the outlet and consequently the outflow becomes activated. However, shortly after the end of the wet season it can be assumed that the lake level is approximately equal to the elevation of the outlet. This level can be seen as the maximum storage level and therefore the initial water level at the beginning of the dry-season. The level may change because of geomorphological changes in the outflow river bed at scales that are however too fine to resolve with our sediment balance approach. No direct measurements of the outlet

elevation are available except for 2019, where the minimum elevation of the outflow channel was 329.8 m asl. The initial lake level for each hydrological year is therefore approximated by the lake level determined for October 1st of the homogenized WSH time series (see Section 4.6.1). The Sen's slope test is then applied to assess the changes in initial lake levels ($\Delta h_{ini}$).

### 4.6.3 Timing of the wet season

To determine the beginning and the end of the wet season we calculate weekly totals of open-water evaporation and compare

them with the precipitation sum of each calendar week. The beginning of the wet season is then defined as the first week in the





year where the weekly precipitation sum exceeds the weekly evaporation. Accordingly, the dry season starts the week after the last week in the year where weekly P exceeds weekly ET. Changes in the DOY of the dry season onset are assessed by applying the Sen's slope test. To translate these changes into millimeters of water ($\Delta h_{\Delta t}$), the Sen estimate of the slope is multiplied by the multi-annual mean daily lake level change calculated based on the average $\Delta h$ seasonality (see Section 4.6.1).

If the lake level decreases below 328.64 m asl the lake level cannot be determined because the bathymetry below that level is unknown. For some years we therefore have data gaps in the WSH time series in late May and June. For this reason, the effect of changes in the timing of the onset of the wet season on water levels is not assessed quantitatively. Because such data gaps rarely occur before May 15th, only WSH data points until that day are considered for assessing the change in end-of-dry season lake levels.

## 5    Results

### 5.1    Validation of gridded datasets

The following variables from the gridded datasets can be validated against station data from Guioyo: precipitation, air temperature and relative humidity (Figure 2). Precipitation and relative humidity reveal a strong seasonality with a peak around August. From November to April precipitation is practically zero. Temperature is relatively constant throughout most of the year (July to January), with slightly higher values towards the end of the dry season (April/May). The gridded datasets reproduce this seasonality very well (Figure 2). Also in terms of absolute differences the gridded datasets agree well with the station data: the mean monthly absolute difference between the ensemble of gridded datasets and the measurements is only 0.6°C (temperature), 12 mm (precipitation) and 8% (relative humidity), respectively.

### 5.2    Lake water areas

In total, 541 satellite images from 527 days within the period 1 October 1999 to 30 June 2021 fulfill the quality criteria and are available to extract WSH data (Figure 4a). The number of available scenes per year increases strongly in the year 2013, when Landsat 8 was launched, and again in 2016, when Sentinel-2 data became available. The minimum number of scenes available per year is 4 (years 2001 and 2004), whereas the maximum is 70 (year 2020).

The extracted lake areas reveal a strong seasonality in lake extent, ranging between 1 ha and 157 ha (Figure 4a). The lake has never fallen entirely dry in the study period (de La Rocque and Renoullin, 2015).

81.9 % of the 10 m pixels within the maximum lake extent area represented the shoreline in at least one scene over the entire study period. For 5.8 % of the area no data are available because of data gaps in the DEM. The remaining 12.3 % of the area either never represented the shoreline or were masked from the DEM because of adjoining canopies below which the shoreline could not be seen.

Earth **Surface**
**Dynamics**
Discussions

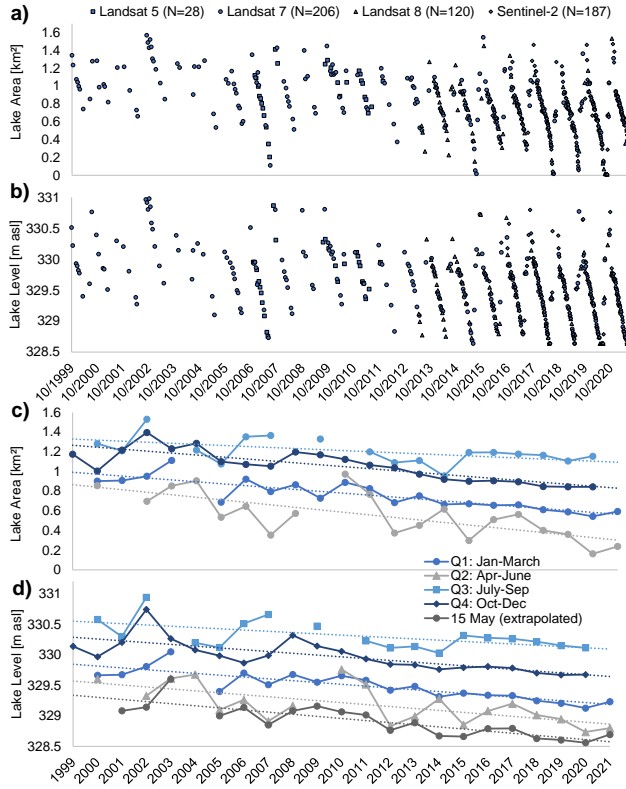

**Figure 4.** Time Series of a) lake area and b) lake water level retrieved from Landsat 5/7/8 and Sentinel-2 optical satellite imagery for the period 1 October 1999 until 30 June 2021. N is the number of scenes available from each sensor. In total 541 scenes are represented. c) Annual average quarterly lake areas and b) lake levels and associated linear trend lines. Lake levels from 15 May in (d) represent the level at the approximate end of the dry season and are identified based on extrapolated lake levels (Figure 10).

## 5.3 Sediment deposition and erosion areas

The iterative approach to identify shoreline elevation anomalies converged to stable solutions after about five iterations (Figure 5). After 10 iterations, the fraction of the area where absolute ΔSEA does not exceed 10 $mm/year$ converged to about 67%. This means that 33% of the pixels for which ΔSEA estimates are available represent deposition or erosion areas that needed to be masked for obtaining the final WSHs. The sediment balance 2000-2021 of the lake bed when WSH is at outlfow level

(329.8 m) converged to a value of -44 mm (Figure 5). The negative sign means that there is more erosion than deposition in the lake bed. The sediment balance could also be determined for some areas that are presently located above the present outflow level - because they were at a lower level in the past or because they represented the shoreline during the wet season when the lake level rose above the outflow level. Here, the sediment balance was +44 mm, which means that there is more deposition than erosion. Overall, the average of all pixels where the point sediment balance could be determined is -12 mm in 21 years.





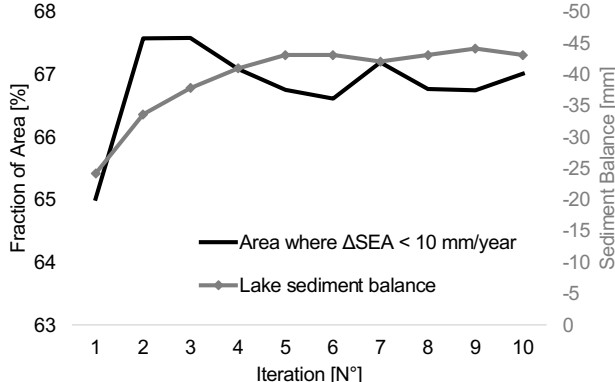

**Figure 5.** Lake sediment balance for the period 2000-2021 (average ∆SEA within the lake bed) as well as the fraction of the area where absolute ∆SEA does not exceed 10 $mm/year$, as a function of the iteration number.

Areas where we observe erosion are distributed over a larger area (20% of all pixels with a valid result) than the areas where we observe deposition (13%). The average net sediment deposition rate at locations with more deposition than erosion is 28 mm/year, whereas at locations with net erosion the average rate is -16 mm/year. Net sediment deposition is concentrated at a few locations such as the river deltas of the southern and eastern tributaries (Figure 3c). The highest average deposition rates are identified at the western shore of the lake, where the southern tributary flows into the lake (up to +62 mm/year). The erosion areas, on the other hand, are stretched along nearly the entire lake shore (Figure 3c), whereas the average erosion rates per pixel never exceed -20 mm/year.

### 5.4 Lake water surface heights

The maximum WSH of the entire study period is 330.98 m asl, reached on October 17, 2002. Assuming that the minimum level was not much below the minimum detectable WSH of 328.64 m asl, the lake levels vary within a range of not more than 2.5 meters (Figure 4b).

The annual average quarterly lake levels reveal negative trends across all seasons (Figure 4d). All trends are significant at the 0.01 level except the Q3 trend (July-September; p-value = 0.028). The Sen slope of the linear trend lines indicates an average WSH decrease between -0.46 m (Q3) and -0.71 m (Q2: April-June) over the 22-year study period.

20 out of the 541 scenes represent a lake extent that was equal or smaller than on 9/10 May 2019. Such data points representing a lake level below 328.64 m asl still needed to be considered for the calculation of quarterly average lake levels. For the sake of simplicity a WSH of 328.64 m asl was assigned to such scenes.

More than one satellite image is available for 14 days, and from another 43 days satellite imagery are available from the subsequent day. Excluding two days where the exact water level could not be determined because of the holes in the DEM, we obtain in total 55 pairs of scenes suitable for analyzing the WSH error. Overall, the absolute WSH errors vary between 0.2 and





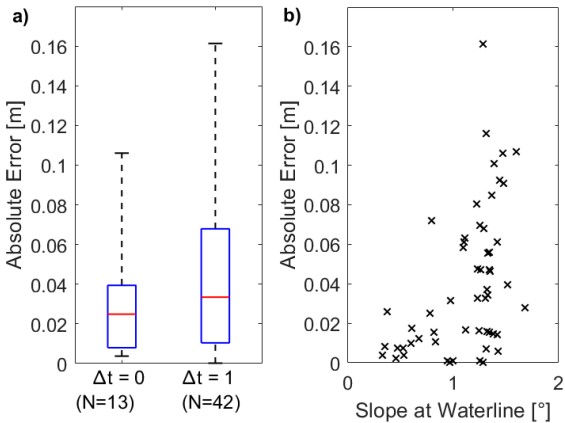

**Figure 6.** Analysis of the relative error between Water Surface Heights (WSH) extracted from Landsat 7/8 scenes and Sentinel-2 scenes. a) Box-plot of the absolute WSH error between scenes of the same ($\Delta t = 0$ days) and of subsequent days ($\Delta t = 1$ day). N is the number of scene pairs available for comparison. The central mark of each box is the median, the edges are the 25th and 75th percentiles, the whiskers extend to the most extreme data points. b) Average slopes of waterline pixels and absolute WSH error between scenes of the same or subsequent days ($\Delta t \leq 1$ day).

161 mm. The median absolute WSH error between scenes from the same day ($\Delta t = 0$ days, N = 13) is 25 mm, and 33 mm if the scenes are from subsequent days ($\Delta t = 1$, N = 42; Figure 6a). If the slope at the waterline was less or equal than $1°$, the median absolute WSH error is only 15 mm ($\Delta t \leq 1$, N = 17), and 51 mm if the slope was above $1°$ ($\Delta t \leq 1$, N = 38).

## 5.5 Water balance components

Of the four water balance components, evaporation rates ($E$) and daily WSH changes ($\Delta h_{rate}$) show the most pronounced seasonality over the course of the dry season (Figure 7). Daily rates of $E$ increase from about 7 $mm/d$ in October to about 10 $mm/d$ in April. $\Delta h_{rate}$ shows a similar behaviour, but the rates are about 2 $mm/d$ lower than the daily rates of $E$. Because $P$ is usually zero during the dry season, the difference between $E$ and $\Delta h_{rate}$ is made up by net inflow ($Q$). Net inflow is positive, which means that inflow is higher than outflow. Only at the beginning of the dry season, in October, outflow from the lake might be still higher than inflow ($Q < 0 mm/d$), but the uncertainty in the calculated rates is high (Figure 7).

The average seasonality of $\Delta h_{rate}$ that is described above is used to fill data gaps in the WSH time series and therefore enabling us to calculate average rates for the dry season of each hydrological year 2000 to 2021 (Figure 8). Only in two years (2000 and 2004) the available observations cover less than 50% of the dry season, and WSHs of these years were therefore not extrapolated. The Sen's slopes of the remaining 19 annual values indicate a positive trend in daily rates of $E$ and $\Delta h_{rate}$, a negative trend in $Q$ and no trend in $P$ (Figure 8).



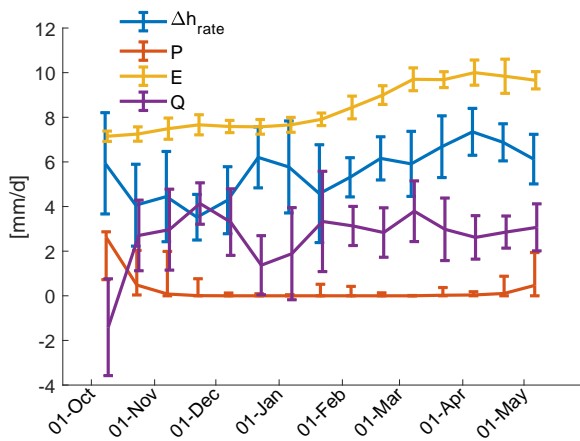

**Figure 7.** Rates of water balance components (lake level change: $\Delta h_{rate}$, net inflow: $Q$, precipitation: $P$, evaporation: $E$) during the dry season (1 October until 15 May). The lines and error bars indicate the mean and standard deviation, respectively, over available data points for 15-day periods. For precipitation the median and the 95% confidence interval is shown instead of the mean and the standard deviation.

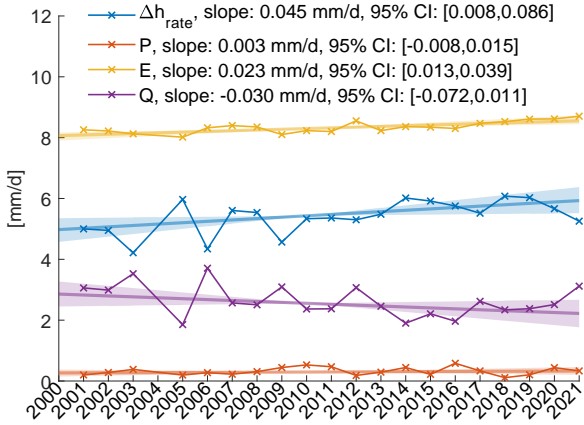

**Figure 8.** Average daily rates of water balance components estimated for the dry-seasons 2000-2021 (lake level change: $\Delta h_{rate}$, net inflow: $Q$, precipitation: $P$, evaporation: $E$). The transparent areas show the 95% CI of the linear regression slope.

## 5.6 Attribution analysis

The increasing trend in dry-season WSH change rates lead to decreasing lake levels at the end of the dry season. Over the study period 2000-2021 the observed change in $\Delta h_{rate}$ cause a decrease in end-of-dry-season lake levels between 37 mm and 406 mm (Figure 9), considering the 95% confidence interval (CI) of the Sen slope of the linear regression (Figure 8). The increase in $\Delta h_{rate}$ can be explained by the increase in direct evaporation from the lake, explaining 62 to 186 mm of the observed





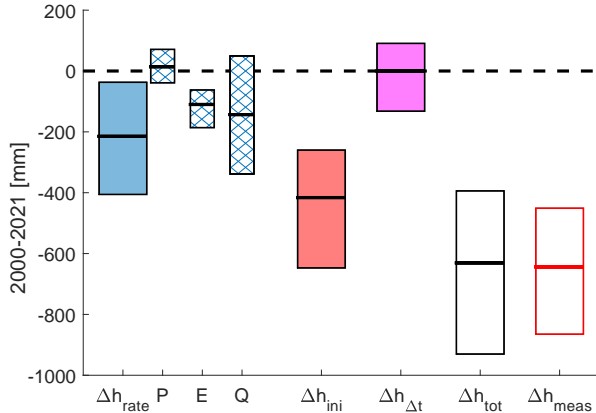

**Figure 9.** Impacts of hydrological changes on end-of-dry-season lake levels (95% CI). $\Delta h_{rate}$ represents the increasing rate of WSH decline during the dry-season that is due to changes in precipitation ($P$), open-water evaporation ($E$) and net inflow ($Q$). $\Delta h_{ini}$ represents the impact of lower initial lake levels at the beginning of the dry-season and $\Delta h_{\Delta t}$ represents the lake level change due to changes in the timing of the wet-season. $\Delta h_{tot}$ is the sum of $\Delta h_{rate}$, $\Delta h_{ini}$ and $\Delta h_{\Delta t}$. $\Delta h_{meas}$ is the measured difference in dry-season lake levels between 2000 and 2021 ($Q1$ in Figure 4).

negative trend in lake levels. Likely, also a decrease in net inflow contributed to a decrease in WSHs. However, according to the test for a monotonic trend, there is a probability of 14% that net inflow has increased rather than decreased. The 95% CI ranges from an impact on the lake level between +50 mm to -338 mm that is due to changes in net inflow (Figure 9).

     The observed lake level decrease over the 22-year period is much higher than what can be explained by the changes in dry-season water balance components. The average lake levels of the months January-March decreased by 644 mm (Figure

4c and Figure 9, 95% CI: 451-865 mm). The extrapolated lake levels for May 15th (Figure 10a) decreased by 414-845 mm (Figure 4c). Even the average lake levels of the wet season decreased by 458 mm ($Q3$ in Figure 4c), but the uncertainty of the regression is high (95% CI: 33-853 mm). Variations in the timing of the dry-season can also not explain the observed lake level trends (Figure 11).The test for a monotonic trend does not indicate that an earlier or later end of the wet season has a significant impact on the multi-year trend in WSH ($\Delta h_{\Delta t}$ in Figure 9).

In spite of an increasing trend in wet season precipitation (ensemble-mean of all precipitation products; see Table 1), the initial lake levels at the beginning of the dry season have been decreasing over time (Figure 10). The decreasing initial lake levels explain 260-647 mm of the observed overall dry-season lake level decrease (95% CI, $\Delta h_{ini}$ in Figure 9).

     The sum of the Sen's slope for $\Delta h_{ini}$, $\Delta h_{rate}$ and $\Delta h_{\Delta t}$ results in a reconstructed lake level decrease of 631 mm (95% CI: 377-958 mm, $\Delta h_{tot}$ in Figure 9), which is very similar to the measured lake level decrease (644±207 mm, $\Delta h_{meas}$ in Figure

9). Changes in $\Delta h_{rate}$ explain about 34±18% of the decrease, $\Delta h_{\Delta t}$ explains 0±11%, and the changes in initial lake levels explains the largest portion of the reconstructed dry-season lake level changes (66±18%).





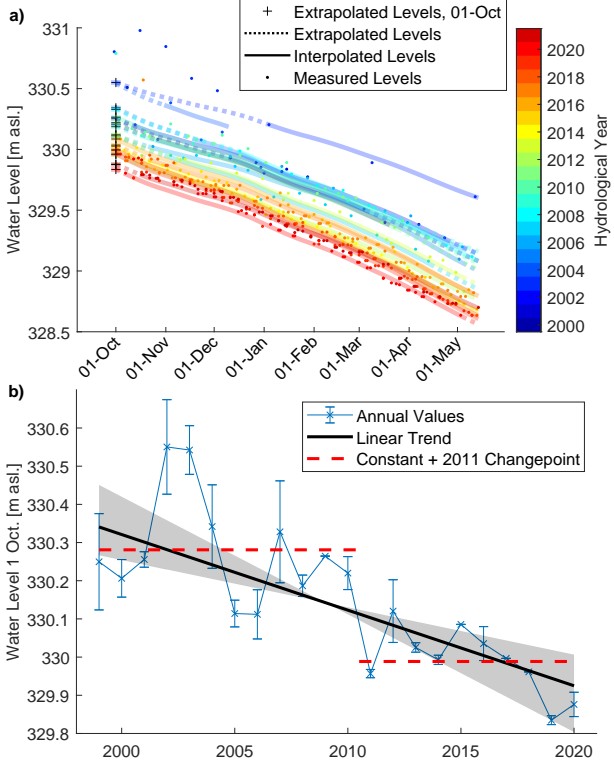

**Figure 10.** a) Dry-season water levels for each hydrological year 2000-2021. The crosses indicate the extrapolated WSH on October 1st, which is used as a proxy for the floor level at the outflow. Values above 330.4 m asl are considered as outliers and are not considered for extrapolation. b) Time-series of extrapolated WSH on October 1st. Error bars represent the range of values obtained by considering the uncertainty in the rates used for extrapolation (Figure 7). The transparent area indicates the 95% CI of the linear regression slope. The dotted red line shows an alternative interpretation of the data, speculating that a single strong erosive event in 2011 may have lowered the outflow level by approximately 29 cm.

# 6 Discussion

The careful separation of the lake sediment and the lake water balance allows us to unambiguously attribute causes to the observed trend of an ever more diminishing lake surface (Figure 4c). The average over all grid points is negative (-43±1 mm
over the period 2000-2021), which means that there is more erosion than deposition, and therefore silting is not the cause of the observed lake area decrease (minus 22-54 ha across all seasons, see Figure 4c). Still, as expected, net sediment deposition is evident at the mouths of the two tributaries. With the derived sedimentation and erosion rates for each pixel we can reconstruct the bathymetry of Lac Wégnia from the beginning of the century, and compare the lake areas for given WSHs. If the WSH is between 329.4 and 329.9 m asl., the lake area today is up to 4.5 ha smaller than 21 years ago (Figure 12a), which represents
about 5% area loss.





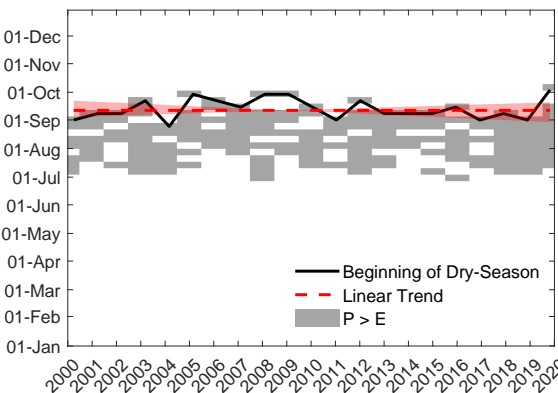

**Figure 11.** Date of the beginning of the dry-season based on gauge-corrected satellite data (black line). The dry-season is defined as the portion of the year where open-water evaporation (E) exceeds precipitation. Weeks with sufficient precipitation to satisfy E are marked in grey. The dotted red line indicates the linear regression line based on the Sen's slope estimator.

At an elevation range between 328.7 and 329.3 m asl. the lake has seen a net erosion of the lake bed (Figure 12). The areas which see a net erosion of the ground are characterized by gentle slopes and are frequently visited by livestock for watering. The bare silty soil dries out during the dry season, gets mobilized by the cattle and then gets suspended in the water when the water is rising again. Our calculations cannot clarify if from there the sediments are moved to deeper areas of the lake where

they deposit again, or if they get flushed away during the wet season. Given the fine grain size of the material we suspect that the latter is the case.

The attribution analysis has revealed that the main cause of the decreasing dry-season water level trend are lower initial lake levels at the beginning of the dry season (Figure 9. The initial lake levels are crucial for the persistence of the lake during the dry season, because the lake does not see any surface water inflow over a period of about eight months. Less than 30%

of the water that evaporates is replaced by net inflow through groundwater exchange (Figure 8). Decreasing initial lake levels therefore imperatively lead to decreasing WSHs at the end of the dry season. The lake might even run completely dry in the future, which is something that has never occurred in the past decades.

The lake levels at the end of the wet season mainly depend on the base level at the outflow, which defines the storage volume of the lake. The sediment balance around the location of the outlet indicate net erosion rates of $10\pm13$ mm per year, which

is less than the $20\pm3$ mm per year obtained for $\Delta h_{ini}$ (Figure 9). However, the outflow channel is only about 5 m wide, and therefore below the pixel resolution of the sediment balance map (Figure 3c). A hand-made dam with a height of about 50 cm made out of sand bags was present at the location during field visits in 2019 and 2020 (Figure 13). According to information from local residents, the dam is not able to withstand the speed of the water leaving the lake during the rainy season and thus collapses, but has been rebuilt each year since 2009 to reduce the outflow of water from the lake. As a consequence of the

collapse of the dam, the flow velocities temporarily increase, which may lead to higher erosion rates. The step between 2010



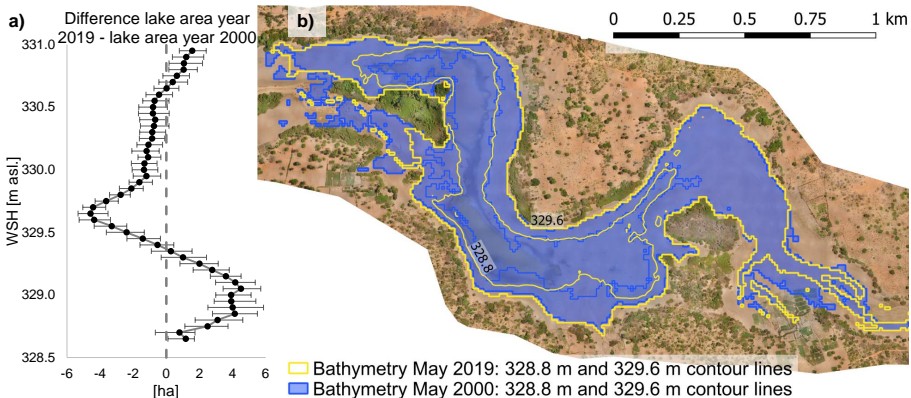

**Figure 12.** a) Difference in lake area in function of water surface height (WSH), based on the bathymetries for May 2019 and May 2000, respectively. The error bars reflect the 95% CI of the sediment erosion and deposition rates that were used to reconstruct the bathymetry of the year 2000. b) Lake contour lines for a WSH of 328.8 m asl and 329.6 m asl, respectively, considering the bathymetries of the year 2000 and 2019.

and 2011 in the time series of the October 1st water level (Figure 10b) coincides with the timing of the first construction of the dam. The average reconstructed water levels are 29 cm higher at the beginning of the dry seasons 1999-2010 than over the period 2011-2020 (Figure 10b). The intervention in the river bed in 2009 may thus have entailed a strong erosive event during the wet season of the year 2011.

A lower outflow level explains also the tendency of lower WSHs during the wet season (Figure 4d), in spite of constant or increasing precipitation (Table 1; confirming recent findings by e.g. Nouaceur and Murarescu (2020) or Bodian et al. (2020)). A lake level of 329.8 m asl. today is 1 m above the outflow level, but about 0.4 m less at the beginning of the century. Consequently, more outflow is generated for the same WSH, and higher inflows are required to reach the same water levels.

The present study also indicates a possible decrease of net inflow to the lake during the dry season, and an increase of direct
evaporation from the lake (Figure 9. Increasing evaporation may be related to increasing air-temperature (Touré Halimatou et al., 2017). Since net inflow is predominantly positive (Figure 7), but there is no surface inflow to the lake during the dryseason, the inflow enters the lake through groundwater exchange. If the groundwater levels are dropping then this contribution is reducing and the lake levels gradually decrease. Groundwater is not continuously monitored in the region, and it is therefore not possible to validate this finding. Our findings highlight the need for the monitoring of groundwater in the region, which has
an important ecological role and which is an important resource for human activities.

## 7   Accuracy assessment

The analysis has benefited from several factors that facilitated the application of the waterline method at the study site:

– Cloud-free meteorological conditions during the dry season.



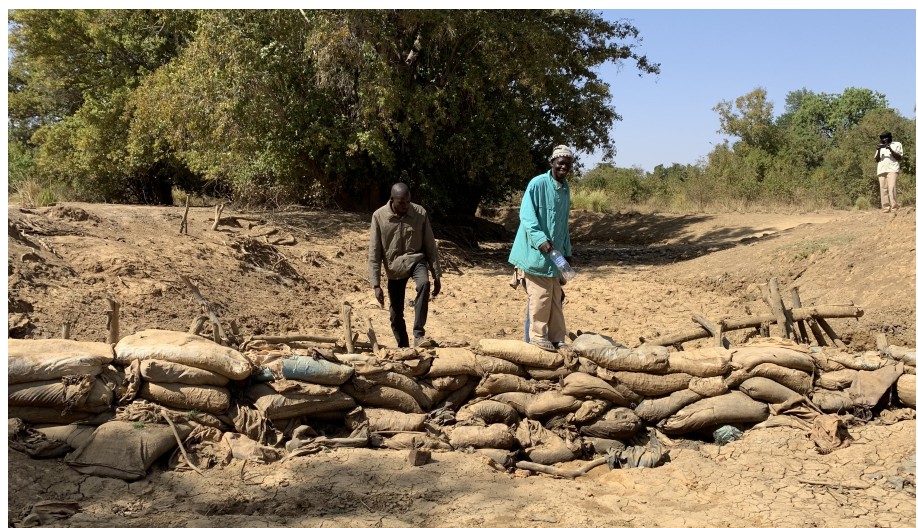

**Figure 13.** Photo of the hand-made dam at the outlet of Lac Wégnia (9 February 2020, Source: Tobias Siegfried).

– Strong natural fluctuations of the water level.

– Shallow slopes at the waterline.

– The availability of a high resolution DEM that also captures the bathymetry of the lake for low water levels.

The climate and geomorphology of the study region is characteristic for large areas in Sub-Saharan West Africa. The first three points are therefore valid for hundreds of water bodies in the Sahel and beyond. The present study can be a showcase for the monitoring of West African lakes using remote sensing, under the condition that the bathymetry is available. Our study has
shown that a single drone survey is sufficient to satisfy this point. However, we are aware that because of the remoteness of the area or for security reasons a visit with a drone is rarely feasible. Satellite altimetry can therefore be a promising alternative for data collection in the field. Armon et al. (2020) have shown that determining the bathymetry of shallow desert lakes using ICESat-2 altimetric tracks is possible. For Lac Wégnia, however, ICESat-2 cross-sections are not yet available for the central part of the lake.

Shallow slopes greatly increase the accuracy of the waterline approach (Tseng et al., 2016). The steeper the slope, the larger the elevation range within a Landsat or Sentinel-2 pixel that represents the lake shore. The average slope of shoreline pixels at Lac Wégnia is between 0.2° and 2.25° per scene, with an average of 1.06° across all scenes that represent a WSH below outflow level (329.8 m asl). Assuming a horizontal error of the waterline position of one pixel (i.e., 30 m for Landsat scenes and 10 m for Sentinel-2 scenes), and a slope of 1.06°, we obtain vertical errors of 0.56 m (Landsat) and 0.18 m (Sentinel-2),
respectively. The average number of 10-m shoreline pixels per scene is n=671 (WSH below 329.8 m asl, excluding masked pixels). Assuming that the vertical error estimates above represents the standard deviation ($\sigma$) of each pixel-wise WSH estimate, we obtain a standard error of the WSH equal to $\sigma/\sqrt{n}$, and therefore 21 mm (Landsat) and 7 mm (Sentinel-2), respectively.



These theoretical errors agree well with the identified relative errors (Figure 6), and demonstrate that the waterline approach provides sufficient accuracy for the purpose of identifying WSHs at Lac Wégnia.

For the shoreline elevation anomalies the same considerations on accuracy apply as for the WSH estimates. The accuracy of the sediment balance estimates then mainly depends on the adequacy of the regression slopes fitted to the SEAs. For individual pixels, the average 1-$\sigma$ CI of the Sen's slope is quite large ($\pm$7.8 mm/year, resulting in an uncertainty of the pixel-wise sediment balance of $\pm$163 mm over 21 years). Sediment balances of individual pixels should therefore be interpreted with care. However, the total number of pixels for which sediment balances could be calculated is n=10,790, and the standard error of the lake sediment balance is therefore only 2 mm. The resulting deposition/erosion patterns (Figure 3) agree well with the expected patterns (see section above), which demonstrates the suitability of the approach.

## 8 Conclusions

This work has demonstrated the utility of the waterline method for extracting the water levels of Lac Wégnia, a Malian lake at the boundary between the Sahelian and the Sudanian eco-climate. 541 WSH data points were obtained for the period October 1999 to June 2021. The data reveal that the lake is dwindling at alarming rates, with a decrease of the seasonal average WSH between 22 $mm/year$ and 33 $mm/year$, which translates to a wet-season area loss of 17% and an end-of-dry-season area decrease of 64% between 2000 and 2021. Based on gridded global datasets and the observed WSH changes we have successfully unravelled the dry-season water balance of the lake. The analysis revealed that a change in the water balance components explains only 34$\pm$18% of the overall lake level decrease, while the reduction of storage volume by erosion of the gully at the outlet of the lake is identified as the main driver (66$\pm$18%) of the observed changes. Future interventions for safeguarding the wetlands of the RAMSAR site should focus on preventing erosion at the outlet channel. In this respect, efforts by local villagers to artificially rise the water table of the lake through improvised dams may be counterproductive, as they increase erosion rates if they are not properly constructed.

The waterline method was further developed in this study to identify shoreline elevation anomalies, which indicate locations with significant sediment erosion or deposition. This novel contribution to the waterline method enables the calculation of sediment balances for pixels that are frequently representing the shoreline. Moreover, accounting for SEAs allows to unambiguously separate lake level and terrain height changes. We could show that no significant silting had taken place within the study period of 21 years. Erosion of the lake shores has even led to an increase of the storage volume by about 15000 $m^3$ below 328.8 m asl. (which is the present floor level at the outflow). When contemplating the sedimentation and erosion of natural lakes, important parallels can be drawn for the planning of reservoirs. The proposed method and the presented results have therefore numerous potential applications.

A remarkable fining of this study is that the lake levels are decreasing in spite of precipitation trends that indicate a possible increase of wet season precipitation. This conclusion does not lack a certain irony, because large areas of the Sahel region saw a surface water increase despite a general precipitation decline during the last decades of the 20th century. While the 'first' Sahelian paradox could be related to large-scale eco-hydrological changes, the paradox reported in this study has its main

cause in the local management of the lake. However, the present study also indicates a possible decrease of net inflow to the lake during the dry season, and an increase of direct evaporation from the lake. Both factors could also negatively impact the persistence of other water bodies in the region. While the present study can be a showcase for monitoring Sahelian lakes using remote sensing, it is hoped that the hydrological trends at Lac Wégnia are not symptomatic for the entire region.

*Data availability.* The pixel sediment balances of Lac Wégnia can be accessed and visualized through an Earth Engine application (https: //hydrosolutions.users.earthengine.app/view/wegnia-sb). This application allows users to click on any point in the lake to view the annual SEAs and the fitted regression slopes. The application also provides access to all available water surface area and WSH data points.

*Author contributions.* TS and RD planned this investigation and are responsible for field observations. SR designed and conceptualized the research, and analysed and interpreted data together with TB. SR wrote the paper with critical review and input from PM. All co-authors
participated in the co-editing of the manuscript.

*Competing interests.* The authors declare that they have no conflict of interest.

*Acknowledgements.* All the participants in the field visits are gratefully acknowledged, namely Moussa Savadogo, Cissé Bocar, Hirosi Pascal Dakouo and Robert Naudascher. We are grateful to Sylvatrop Consulting directed by Sylvain Dufour for carrying out the UAV survey in May 2019. Sylvatrop Consulting also processed the aerial images and generated the DEM. We thank Rebecca Hochreutener and Abdouramane
Yoroté from TAHMO for organizational and technical assistance. This research is supported by the SAWEL program "Improved Food security and nutrition in the Sahel by safeguarding wetlands through ecologically sustainable agricultural water management" carried out by Wetlands International, Caritas Switzerland, the International Water Management Institute and hydrosolutions GmbH, with financial support from the Swiss Agency for Development and Cooperation (SDC).



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
