# Peer review of "Unraveling the hydrology and sediment balance of an ungauged lake in the Sudano-Sahelian region of West Africa using remote sensing"

_Earth Surface Dynamics, 2021_

## Author Comment (AC1)

**Response to 'Comment on esurf-2021-99' from Anonymous Referee #1**

We thank the referee for her/his constructive and detailed review which will help us to improve this paper. Please find all the details below.

+++++++++++++++++++++++

General Comments:

+++++++++++++++++++++++

This study provides several different sets of outcomes, and for the most part they are handled very well. The clearest (yet somewhat less innovative) part of the paper is about modeling water levels in a lake based on a combination of environmental data (precipitation, evaporation) and remotely sensed measurements of lake shoreline position, which in turn are based on overlaying a binary land/water mask on a digital elevation model (DEM). I thought this aspect of the paper was quite convincing. The second part of the work, also remote-sensing focused, was on quantifying sedimentation and erosion, by examining departures from uniform elevation along observed shorelines on different dates. This aspect was innovative and interesting, and the combination of these first two parts makes a nice model for how others might proceed to do similar kinds of analysis in other locations.

The third aspect of this paper was a water budget analysis, based on the other components, from which the authors claim that the observed long-term downward trend in lake water level is due to erosion (deepening) of the outlet channel. That is certainly plausible, but I found the explanation somewhat unclear and unconvincing. Despite obvious increases in water loss during the dry season -- increased evaporation, decreased net inflow, and the resulting more-negative trend in water levels during each dry season -- the authors argue that the lower dry-season water levels can be explained mostly just by lower water levels at the start of the dry season. I have questions about that interpretation, as discussed below.

Particularly on the remote sensing side, I would applaud the authors' thoroughness and attention to small details. There are many points in this analysis where the authors actually go somewhat further than I would normally expect, e.g., the iterative approach used in quantifying shoreline deposition and erosion described at lines 255-261. Overall, the remote sensing analysis appears very comprehensive and well planned. I believe these parts of the manuscript do make a valuable contribution to the literature. Before moving on to my specific comments and technical corrections, I will briefly respond to queries from the journal's peer review criteria: Does the paper address relevant scientific questions within the scope of ESurf?

Yes.

Does the paper present novel concepts, ideas, tools, or data?

Yes, though some parts are more novel than others.

Are substantial conclusions reached?

Yes, many substantial conclusions are included.

Are the scientific methods and assumptions valid and clearly outlined?

For the most part, yes. See comments below.

Are the results sufficient to support the interpretations and conclusions?

The remote-sensing based conclusions seem well justified. I am a bit more skeptical (or confused by) some of the interpretations.

Is the description of experiments and calculations sufficiently complete and precise to allow their reproduction by fellow scientists (traceability of results)?

Yes.

Do the authors give proper credit to related work and clearly indicate their own new/original contribution?

Yes.

Does the title clearly reflect the contents of the paper?

Yes

Does the abstract provide a concise and complete summary?

Yes

Is the overall presentation well structured and clear?

Yes, with the exception that some of the presentations of the results, and of the authors' interpretations, were confusing or unclear to me.

Is the language fluent and precise?

Yes, with minor corrections noted below.

Are mathematical formulae, symbols, abbreviations, and units correctly defined and used?

Yes, with minor corrections noted below.

Should any parts of the paper (text, formulae, figures, tables) be clarified, reduced, combined, or eliminated?

See comments below.

Are the number and quality of references appropriate?

Yes.

Is the amount and quality of supplementary material appropriate?

Yes.

**Response on General Comments:**

The main criticism of the reviewer concerns the interpretation of the results regarding the lake water level downward trend attribution analysis. Based on our analysis, the main cause of this long-term downward trend is the erosion (deepening) of the outlet channel. The reviewer agrees that our conclusions are plausible, but states that the presentation of some of the results and the interpretations were confusing or unclear. We thank the reviewer for pointing this out. We will revise the presentation of the results and we will improve our argument as outlined in the detailed response to the specific comments below.

++++++++++++++++++++++

Specific Comments:

++++++++++++++++++++++

1. I am a bit confused by the attribution process here, and it's probably just my own failure of understanding (but if so, perhaps some things could be written more clearly). The last line of the abstract states unequivocally that lowering the floor level of the outflow (erosion) "explains the decreasing trend in WSH". But as lines 445-450 note, a long-term change decrease in groundwater recharge (perhaps due to regional warming and increasing ET rates in the lake's watershed) could also contribute to the observed reduction in surface height. All the discussion of evaporation in the manuscript appears to focus on evaporation from the lake itself (and even that does appear to explain a significant fraction of the overall change in lake level, as shown in figure 9). Would it be more realistic to say that the declining trend in WSH is explained in part by evaporation from the lake surface (~15% or whatever), and that the remainder is attributed in part to a potential lowering of the outlet channel, but long-term changes in groundwater recharge are unknown? (I have larger concerns about the attribution issue below.... see my comments 2, 7, and 9 below.)

The reviewer is right that the lowering of the floor level of the outflow is not the only cause of the declining trend in WSH, and that the other causes that we have identified (increasing evaporation, potential changes in groundwater recharge) should also be mentioned in the abstract. The fractioning of the contributions to the losses is provided on lines 405-406 of the manuscript. We will reference to these results in the revised abstract.

2. Lines 147-148 say "The analysis focuses only on the loss part of the year to explain the ecologically most critical conditions for the lake." However, the main conclusion seems to be that what matters is the starting point of each dry season (what is the water level at the beginning of the dry season?). But the general downward trend in the "starting point" could in principle come from an increasingly negative water balance during either the wet or dry season. In other words, how do we know whether the problem is bigger losses during the dry season, or smaller gains during the wet season? I understand from lines 148-150 that cloud cover during the wet season prevented assessing the dynamics *within* wet seasons (e.g., time series in lake properties from June to Sept). But looking at the net change in area and water level (volume would be better...) from the start to the end of each season should be possible, I'd think? Basically, what are the long-term trends in (a) the change from Oct 1-May 15; and (b) the change from May 15 to Oct 1? I'm trying to extract that information from Fig 4 and not really succeeding.

The reviewer is raising here the question how we can exclude that wet season lake volume changes are not causing the observed overall downward trend in WSH (through smaller gains during the wet season). Here it is important to note that the lake level raises above the floor level of the outflow every year. The measured floor level at the outflow in May 2019 was 329.8 m asl. This value was exceeded every year by the available wet-season WSH maximum values (330.1 – 331.0 m asl.). The lake can therefore be seen as a reservoir that fills up during the wet season and gradually empties during the dry season. Consequently, the volume change is zero, if calculated from the moment where the outflow gets activated (in June, usually) until shortly after the end of the wet season, when the outflow gets deactivated again. Figure 11

confirms that the wet season always ended before Oct 1$^{st}$. From the moment on where the outflow gets deactivated, WSH start do decline at a relatively steady rate ($\Delta h_{rate}$, see Figure 7). Knowing this rate allows us to reconstruct the water level on Oct 1$^{st}$ (Figure 10b), also in years where no data point is available around that date (Figure 10a), and this reconstructed water level corresponds to the floor level at the outflow.

Given our study goals there is therefore little interest in further examining the wet-season water balance, including the derivation of all components in Eq. 1. This is what we wanted to state by lines 147-150. However, to state that "the wet-season water balance is not investigated in this work" was misleading, and we will correct that in the revised manuscript. We did investigate the behaviour of the lake during the wet season, as outlined above, and considered it for our interpretations and conclusions. We will state this clearly in the revised manuscript.

3. Section 4.4.1: I am not completely convinced by the coregistration explanation; the calculations of erosion and sedimentation would seem to be very sensitive to minor errors in coregistration. Ideally it would be helpful to include a sensitivity analysis of that. I'm open to the argument that the long-term trends visible in Figure 3 suggest that any errors caused by coregistration are random not systematic and are cancelling out over time (i.e., contributing noise rather than bias). But a bit more discussion of the assumptions and effects related to image coregistration is needed.

It is true that the point sediment balances at individual pixels are sensitive to errors in the coregistration. A systematic coregistration error in West-East direction would lead to a positive error in $\Delta SEA$ on one shore and to a negative error on the opposite shore. However, we do not observe such positive/negative patterns at opposite sides of the lake in our case, as the reviewer has noted as well. According to Nguyen et al. (2020), the random offset differences between Landsat 8 and Sentinel-2 can be reduced to "less than 2 m in most of the pixels" with the displacement() function in GEE (which is the function that we are using for co-registration). To test the sensitivity of the lake sediment balance to a random error in coregistration, we have added a random error with a normal distribution ($\mu=0$, $\sigma=2$ m) to the determined displacements (both in x and y direction). The resulting lake sediment balance over the 22 years study period changed from -44 mm to -35 mm (+9 mm). We will add this discussion to the revised manuscript (Section 4.4.1 and Section 7).

4. I really had trouble interpreting the image with water levels in Figure 1. My initial understanding was that the large gray patch was the actual image of the lake in the orthoimagery, when it was at 328.7 m. Only later did I figure out that this is a partiallytransparent gray polygon superimposed on the orthoimage to represent the area between 328.7 m and 330 m (... at least, I think this is correct) and that the areas below 328.7 m are the two large "island-like" patches outlined in black, one of which has a greenish color and one a more tan color. I initially perceived those as actual islands sticking up above the 328.7 meter contour, when they're actually pools that extend below it. To make this map clearer, consider instead using two different and contrasting colors for outlining the 330 and 328.7 meter contours, or use two different opaque fills (like dark and light blue, or dark and light

gray) for areas below 328.7 and from 328.7 to 330 m. This is a problem because I initially assumed there were no elevation values available within the entire gray area.

Please apologize, we realize that the legend in Figure 1a was misleading. Indeed, the water area at level 330 m is represented by a semi-transparent area, whereas the area at level 328.7 is represented by a thick black line. The patches of muddy water remaining at a level of 328.7 m may therefore get misinterpreted as islands, which they are not. In the revised manuscript we will improve Figure 1 as suggested by the reviewer, to prevent such misinterpretations.

5. Figure 4: The pattern of colors and shapes is inconsistent between (c) and (d) – I assume they should be ordered the same (top to bottom), but for clarity, use the same colors and shapes for the lines and point symbols. The legend starts in panel (c) and extends onto panel (d) - maybe move it to the bottom of the figure, or between (c) and (d), with a horizontal rather than vertical layout?

The reviewer is right that the pattern of shapes was inconsistent between Figure 4c and Figure 4d. Thanks for pointing this out. We will correct his and also reconsider the position of the legend.

6. Figures 7 and 8: The values for dh(rate) are shown as positive. I assume they are actually negative (decreasing height). Perhaps it might make sense to label them all as components of change in height, i.e. dh(E), dh(P), dh(Q), dh(Total) with dh(E) and dh(Total) being negative, and dh(P) and dh(Q) being positive? Likewise, in line 389 (bottom of page) it is confusing to say that an increase in evaporation caused an increase in dh, when dh is becoming more negative. Finally, in Figure 8's legend, there are values given for the slopes that are the same units as the Y axis itself. Should the slopes be in mm d-1 y-1 (mm/day/yr)?

Thank you for the careful inspection of Figures 7 and 8. We agree with the reviewer here. We showed all variables with positive values so that they can be plotted close to each other. However, we will revise the figures and the text so that the variables are presented with correct signs, units, and labels.

7. Lines 393 and following return to the attribution question, and I am still confused and failing to follow the authors' argument. They write "The observed lake level decrease over the 22-year period is much higher than what can be explained by the changes in dryseason water balance components. The average lake levels of the months January-March decreased by 644 mm..." But 644 mm over 22 years is 29 mm per year. Meanwhile, the daily rate of dry-season change in water level (dh) according to the blue line in Figure 8 changed by about 1 mm/day (from 5 mm/day to 6 mm/day) over the same period. If there are 200 days in the dry season (an underestimate), then 1 mm/day = 200 mm/season. If the lake is losing an extra 200+ mm more each dry season than it did in 2000, why would that be unable to explain a 644 mm change in water level? It would only take 3 seasons with the extra 200mm of water loss to add up to 600 mm total water loss.

The reviewer is not right here, the daily rate of evaporation did not change by 1 mm/day per season, which would indeed result in loosing an extra 200 mm each dry season, but by 0.023 mm/day/year * 21 years = 0.49 mm/day over the entire study

period. Therefore, the effect on the end-of-season lake levels can be quantified as 0.49 mm/day * 225 days = 110 mm. Figures 8 and 9 present the results correctly, although with wrong units for the slope of ET, as pointed out by the reviewer in the comment above. We will correct this in Figure 8, and we will also try to explain this better in the revised manuscript.

8. Lines 411-412: The authors write "Still, as expected, net sediment deposition is evident at the mouths of the two tributaries." But in figure 3c, it looks like there is net erosion at the extreme east arm of the lake, where the eastern tributary enters (I believe). There is net deposition (red) slightly further along, but where the tributary actually enters it looks more like erosion (blue).

The sediment deposition is best visible at the deposition plains downstream of the mouths of the two tributaries, and not at the mouths itself. We will state this correctly in the revised manuscript.

9. Line 422: See above comments about attribution. I am still having trouble following the authors' argument that lake levels are getting lower at the end of the dry season because they're starting out lower at the beginning of the dry season, and that changes during the dry season itself are unimportant. But if all else (wet season contribution) stays the same, and slightly more water is lost each dry season, there will be a downward trend in the starting levels for each dry season, purely due to the increasingly negative balance in the previous dry seasons. In oher words, the authors appear to be saying the cause of the long-term decline is not A (what happens during dry season) but B (starting point for the dry season) ... but A in all previous seasons affects B in all subsequent seasons. I am probably misunderstanding things, but I just don't think the authors are demonstrating the argument for attribution very clearly here.

As we have stated under comment 2, the lake reservoir entirely fills up during every wet season. Therefore, the long-term decline in water levels at the end of the dry season does not affect the water levels at the beginning of the dry season. It may affect the total runoff from the lake during the wet season, though. Possibly, the outflow from the lake is nowadays delayed because the lake reservoir is almost empty at the end of the dry season and needs to fill up first. However, the focus of the paper is not on quantifying runoff from the lake during the wet season, so we have not tried to quantify such effects. Overall, we believe that our interpretations and conclusions regarding the long-term downward trend in dry season water levels are robust and the attribution analysis considers the most relevant factors.

We will try to improve the clarity of our explanations in the revised manuscript. Certainly, we do not want to create the impression that changes during the dry season are unimportant. We will emphasize the role of evaporation and groundwater exchange by mentioning this in the abstract, as stated below comment 1 above.

10. Section 7 is labeled "Accuracy Assessment", but the firrst part (through line 464) is not really an accuracy assessment; it's a discussion of how generalizable the methods used here are for other sites

elsewhere. More generally, in the remote sensing community this phrase "accuracy assessment" usually involves some comparison with a second, external dataset that's used to validate estimates produced from the first dataset. In this case, both the original estimates of uncertainty, and the calculated ones in this section (465-481) are both derived from the same source - the UAV's DEM produced in Photoscan. Errors in that DEM would affect both the "results" and this "accuracy assessment" ... in other words, the accuracy assessment isn't really independent. Consider just treating this section as a continuation of the discussion? Or move the first part (through line 464) into the discussion, and name Section 7 "Uncertainty Estimates" or something like that?

The reviewer is right that an independent dataset is not available for accuracy assessment. We will move part of the text in Section 7 to Section 6 (Discussion) and rename Section 7 to "Uncertainty assessment".

+++++++++++++++++++++

Technical Corrections

+++++++++++++++++++++

11. In general, there should be no hyphen in "wet season" when it's not used as a modifier. So remove it in line 12, in line 278, and in the caption to Figure 9. It's OK to keep it hyphenated at line 148 (modifying "water balance") and 486 (modifying "area loss"). Likewise, for "dry season", remove the hyphen when it's not being used as a modifier (e.g., remove it in line 277, but keep the hyphen in line 279).

Ok

12. The sentence extending from lines 78-79 is a bit garbled at the end ("or for Water volume estimates of desert lakes Armon et al. (2020)"). Perhaps the authors mean something like "...Xu et al., 2020), or for water volume estimates of desert lakes (Armon et al., 2020)."

Yes, the reviewer is right. We will correct this.

13. Line 86: I would delete the word "within" ("... at the lake shores and the lake bed...")

Ok

14. Figure 1 caption: I would say "Orthoimage mosaic from UAV on 9/10 May 2019 [...] and on the days of the UAV flights ..." In general, the authors use "UAV" more frequently than "drone", and I would recommend being consistent in usage (replace other uses of "drone" with "UAV" or "drone survey" with "aerial survey" in lines 114, 460, and 461.

Ok, we will do this.

15. Line 106: Maybe replace "effluent" with "outlet"

Ok

16. Line 128: There is an unclosed parenthesis before "Section 4.2: GLDAS..."

Thanks, we will correct this.

17. Line 138 (last line on page 5): should be "were therefore also..."

Ok

18. Line 147: Add space in "dryseasons"

Ok

19. Line 180 and 182: "orthoimage"

Ok

20. Line 245: This sentence (starting with "Not" on the previous line) is rather confusingly worded. Consider something like this: "While individual dates' SEA values may represent noise, the longer-term trend in SEA values (dSEA, units mm/year) point to deposition and erosion processes."

We will reword the sentence as suggested.

21. Line 300: Add space in "wetseason"

Ok

22. Line 344: Replace "outfolw" with "outflow"

Ok

23. Lines 386 and 387: Use either "leads to" and "causes a" or "led to" and "caused a"

Ok, we will correct this.

24. Line 492: "raise" not "rise"

Ok

25. Line 502: "finding" not "fining"

Ok

**Response to 'Comment on esurf-2021-99' from Mark F. Muller**

The paper presents a (in my opinion clever) technique to simultaneously assess changes in water elevation and sediment balance based on changes in the position of the shoreline monitored with high resolution satellite imagery. The approach crucially relies on a high resolution DEM obtained from a UAV. This limits the scalability and applicability of the approach but is duly discussed in the paper. Although focusing on a specific site, the authors make a convincing argument in the introduction to support the broader applicability of both their findings and the developed methodology. As such, I believe that the paper is of high potential interest to the broad readership of ESD. The paper is well organized and well written and, while the process-interpretation issues brought up by the other reviewer are valid and should definitely be discussed and elucidated, I find the methodology itself novel and compelling enough to recommend the paper's prompt publication in ESD.

One (very minor) comment that I have relates to the limitations of the approach. The authors present a solid discussion in Section 7 about the practical constraints/requirements of the approach, but one constraint that is perhaps missing relates to the resolution of the satellite imagery that limits the application of the approach to sufficiently large lakse, whereas smaller ponds and reservoirs are arguably at least as critical to ecological and socio-economic applications in arid regions. This could be easily addressed, for example by extending the discussion on L 465-470 to discuss the number of pixels (n) necessary for the standard errors on WSH and sediment balances to remain "acceptable".

My only other suggestion would be to make the finalized GEE code available (via a static link in the paper) if possible. Many researchers would without a doubt benefit from using this approach and this would increase the impact of the paper.

Again, these are very minor comments. This is honestly one of the most compelling initial submissions that I have reviewed for a long time and I congratulate the authors on their work.

Marc F. Muller

We thank the referee for his positive feedback and his recommendation for prompt publication in ESD.

We have one remark regarding the statement in the first paragraph above that the "approach crucially relies on a high resolution DEM obtained from a UAV". We would like to point out that other authors have shown that the bathymetry of temporary lakes can also be derived from satellite altimetry (Armon et al., 2020). It is therefore possible to apply our approach also based on publicly available data from e.g. ICESat-2 satellite. We state this in the manuscript on lines 461-463. We have also already started to implement such a methodology. Our preliminary results are publicly available through a GEE web-app:
https://hydrosolutions.users.earthengine.app/view/sahel-water

We will follow the reviewers' recommendation to discuss the minimum size of water bodies that can be assessed with our approach, given the typical pixel size and the identified standard errors on WSH.

We will also follow the suggestion to make the GEE codes available. We will prepare a GitHub page for this purpose.

---

## Author Response (AR2)

**Comments to the author by the Associate Editor Richard Gloaguen**

First I commend the authors for the attention to details and their careful implementation of the reviewers comments.
Secondly, I apologize for the delays. It was extremely difficult to find experts willing to assess this submission. I can not find any reason for this. Fact is: I can not further delay the processing of this submission.

I have 2 remaining issues that I would like the authors clarify.

1- I can not understand why ground water levels can be discarded as culprit here. It seems that this a very complex situation and in many lake regions, groundwater flows have a huge impact on lake levels. Is there a way to assess groundwater levels in the region (wells?) .

2 - The conclusions are very regional. It would be important to make statements on 1- the portability of the method to other regions (assess the limitations) and 2- If this single lake is representative of the situation in Sahel.

**Report by Anonymous Referee #3**

This paper extensively presents a very useful method to infer area, height, erosion/deposition balance of an African lake. All data and methods are well presented and discussed in depth. The paper underwent a first round of reviews, which very carefully went through the text, figures and legends (especially reviewer 1). As a result, the revised paper has been corrected form what appeared to me as minor errors/typos. Both reviewers and authors should be commended for this. Some interpretations are closer to hypotheses than to firm conclusions (the lake bottom is not observed), but this is completely normal (discussion) and the paper fully reaches its objectives of presenting a widely applicable method, drawing relevant conclusions and pointing possibly important impacts.
Not being a native english speaking person, I will not comment the language, which seems to be more than OK to me.
Therefore, I recommend to publish the paper as is.

**Response by the authors**

We are grateful to the Associate Editor Richard Gloaguen for his decision to publish the paper subject to minor revisions. We also thank Anonymous Referee #3 for his/her suggestion to publish the manuscript as is, and for commending the "*relevant conclusions*" and "*possibly important impacts*" of the study.

The two remaining issues raised by the Associate Editor we have addressed as follows:

1- Groundwater levels are not discarded as a culprit per-se, and the water balance indicates a possible decrease of net inflow to the lake during the dry season (Figure 9). Inflow during the dry season happens exclusively through groundwater exchange. Falling groundwater could therefore have contributed to the lake level decline. However, other factors such as increasing evaporation from the lake and in particular the decrease of the base level at the outflow have a more significant negative impact on the lake water levels. The uncertainty range of the calculated net inflows indicates that groundwater levels could also be stable. Data from wells are not available in the region. We have therefore endeavored to assess groundwater level trends by using as proxy the surface water extent of Lac Kononi, a small groundwater-fed lake in the vicinity of Lac Wégnia. For this lake there is no

evidence of declining lake areas over the last 22 years (now shown by Figure A1). We think that with this additional analysis and given the results of the water balance we can convincingly show that groundwater levels are indeed not the main culprit here. The additional analysis is discussed on lines 457 to 564 of the revised manuscript. The following text has been added:

*"However, the uncertainty range of the calculated net inflow indicates that groundwater levels could also be stable (Figure 9). Indeed, a general decrease of groundwater levels should also affect the levels and extent of other water bodies in the area, but the surface water areas of Lac Kononi in the vicinity of Lac Wégnia do not show a decreasing trend over the last 22 years (Figure A1). Lac Kononi is a groundwater fed lake that is not directly connected to a river channel, and its water level can therefore be used as a proxy for local groundwater levels. Unfortunately, the bathymetry of Lac Kononi is not available, and we could therefore not reconstruct its WSH such as we have done for La Wégnia. Its constant lake areas, however, are an indicator for stable groundwater levels."*

2. We have revised the Conclusion section to address the portability of the method and the representativeness of our results for the situation in the Sahel. Regarding the portability of the method we have added the following sentences (lines 524-528):

*"The method is portable to all water bodies with strong fluctuations of the water level over periods where optical satellite imagery are available. In case of persistent cloud-cover, the method could be extended to surface water mapping using Synthetic Aperture Radar data (e.g. Markert et al. 2020). No in-situ measurements are required to apply the method, provided that a high-resolution DEM of the bathymetry is available."*

Please note that a more detailed discussion of the portability and limitations of the method is provided by Section 7 (lines 491 - 504).

Regarding the representativeness of our results for the situation in the Sahel, we now clearly state that the decreasing lake level trends cannot be taken as representative of the situation in the Sahel, but the main cause is the local management of the lake. On the other hand, we have identified several issues that are also highly relevant elsewhere in the Sahel (dynamic sediment movements, increasing direct evaporation from water surfaces, groundwater levels, increasing rainfall). No general conclusions can be drawn regarding the importance of each of these factors for other lakes in the Sahel, but the present study represents a showcase for how to unravel their hydrology and sediment balance given the constraints on in-situ data availability. The corresponding paragraph in the Conclusion section has been revised as follows (lines 533-539):

"The present study also indicates a possible decrease of net inflow to the lake during the dry season, and an increase of direct evaporation from the lake. Both factors could also negatively impact the persistence of other water bodies in the region. *However, a small groundwater-fed lake in the immediate vicinity of Lac Wegnia does not reveal a negative trend in water area, and other water bodies in the Sahel are likely benefiting from the positive rainfall trends. The obtained results should therefore not be taken as representative of the situation in the entire region, but* the present study can be a showcase *for unraveling the hydrology and sediment balance of other* Sahelian lakes using remote sensing."